# Identifying Calmodulin and Calmodulin-like Protein Members in *Canavalia rosea* and Exploring Their Potential Roles in Abiotic Stress Tolerance

**DOI:** 10.3390/ijms252111725

**Published:** 2024-10-31

**Authors:** Qianqian Ding, Zengwang Huang, Zhengfeng Wang, Shuguang Jian, Mei Zhang

**Affiliations:** 1Guangdong Provincial Key Laboratory of Applied Botany & South China Agricultural Plant Molecular Analysis and Genetic Improvement, South China Botanical Garden, Chinese Academy of Sciences, Guangzhou 510650, China; dingqianqian@scbg.ac.cn (Q.D.); huangzengwang23@scbg.ac.cn (Z.H.); wzf@scbg.ac.cn (Z.W.); jiansg@scbg.ac.cn (S.J.); 2University of Chinese Academy of Sciences, Beijing 100039, China; 3CAS Engineering Laboratory for Vegetation Ecosystem Restoration on Islands and Coastal Zones, South China Botanical Garden, Chinese Academy of Sciences, Guangzhou 510650, China; 4Southern Marine Science and Engineering Guangdong Laboratory (Guangzhou), Guangzhou 511458, China

**Keywords:** calmodulin, calmodulin-like protein, ecological adaptability, yeast heterogeneous expression, *Canavalia rosea*

## Abstract

Calmodulins (CaMs) and calmodulin-like proteins (CMLs) belong to families of calcium-sensors that act as calcium ion (Ca^2+^) signal-decoding proteins and regulate downstream target proteins. As a tropical halophyte, *Canavalia rosea* shows great resistance to multiple abiotic stresses, including high salinity/alkalinity, extreme drought, heat, and intense sunlight. However, investigations of calcium ion signal transduction involved in the stress responses of *C. rosea* are limited. The *CaM* and *CML* gene families have been identified and characterized in many other plant species. Nevertheless, there is limited available information about these genes in *C. rosea*. In this study, a bioinformatic analysis, including the gene structures, conserved protein domains, phylogenetic relationships, chromosome distribution, and gene synteny, was comprehensively performed to identify and characterize *CrCaMs* and *CrCMLs*. A spatio-temporal expression assay in different organs and environmental conditions was then conducted using the RNA sequencing technique. Additionally, several *CrCaM* and *CrCML* members were then cloned and functionally characterized using the yeast heterogeneous expression system, and some of them were found to change the tolerance of yeast to heat, salt, alkalinity, and high osmotic stresses. The results of this study provide a foundation for understanding the possible roles of the *CrCaM* and *CrCML* genes, especially for halophyte *C. rosea*’s natural ecological adaptability for its native habitats. This study also provides a theoretical basis for further study of the physiological and biochemical functions of plant *CaM*s and *CML*s that are involved in tolerance to multiple abiotic stresses.

## 1. Introduction

In plants, calcium ions (Ca^2+^) act as secondary messengers, playing vital roles in intracellular signaling during various developmental processes and in response to both biotic and abiotic stimuli [1]. The calcium signal is transferred by a series of Ca^2+^-binding proteins; among them, calmodulin (CaM) and calmodulin-like protein (CML) are major calcium sensors that have been shown to be involved in a wide variety of environmental responses and physiological activities [2,3]. CaM and CMLs comprise EF-hand motifs with helix–loop–helix structures for Ca^2+^-binding, acting as sensors in interpreting encrypted Ca^2+^ signals [4,5,6]. Studies have shown that CaM is one of the most conserved proteins, with four EF-hand domains in all eukaryotes [2,7]. In contrast, CML is relatively less conservative, containing 1–6 degenerated EF-hand domains [8,9], and it only exists in plants and some protists [3,4].

Due to their sessile characteristics, plants are inevitably challenged by multiple biotic invasions and environmental changes under normal or stress conditions. These challenges can cause transient fluctuations in cytosolic Ca^2+^ ([Ca^2+^]_cyt_) levels; then, the changes in Ca^2+^ signatures are decoded by Ca^2+^-binding proteins, including CaM and CML [10]. There are five types of Ca^2+^-binding proteins with EF-hand motifs, including calmodulin (CaMs), calmodulin-like proteins (CMLs), calcium-dependent protein kinase (CDPKs), calcium-dependent protein kinase-related kinase (CRK), and calcineurin B-like proteins (CBLs) [5,9]. CaM is a small well-characterized Ca^2+^ sensor and has multiple functions to respond to different biotic and abiotic stimuli. In addition, it is evolutionarily conserved and possesses good heat and acid stability, but lacks its own catalytic activity [10]. CaM can bind to target proteins and control its target proteins through protein–protein interactions or by changing their gene expressions [10,11]. In contrast, CML has greater variability, and its structure is not so conservative. Additionally, convincing evidence has emerged that this type of protein plays central and highly specific roles in coordinating the environmental responses of plants [8,9].

Plant CaM and CML gene families have been identified and characterized in various species, including Arabidopsis [12], rice [13], and other green lineages from algae to land plants [3,14]. *Canavalia rosea* belongs to the family Leguminosae and is characterized as a pioneer species for tropical coastal saline vegetation construction due to its wide ecological adaptability and potential nitrogen fixation ability [15,16]. *C. rosea* is a common vine species in south China coastal wetlands with excellent tolerance to barren soil and salinity/alkalinity that offers rapid growth for horticultural engineering operations. Moreover, due to the nutritional and medicinal value of its seeds, *C. rosea* has also become an important wild plant resource with latent economic value and ecological significance [15,16]. However, there are no reports on the Ca^2+^-binding protein members and their functions in *C. rosea*, especially regarding their ecological adaptation to extreme adversity. 

Previous studies have summarized the potential roles of plant *CaM*s and *CML*s in responding to various environmental stresses [17,18,19], with extensive research indicating that these genes may play key roles in salt or drought tolerance [20,21,22,23,24,25], or even sensitivities [26,27], and responding to other abiotic stresses, such as heat [17] and freezing [22]. In recent years, some *CaM* and *CML* gene families have been identified in special-habitat plants, including hydrophyte *Saccharina japonica* [28], sacred lotuses [29], halophytes *Paspalums vaginatum* [30], and *Nitraria sibirica* [31]. The related results provide insights into the potential function of these plant *CaM*/*CML* genes and their possible roles in the molecular mechanisms of extreme tolerance to abiotic stresses. 

With *C. rosea* being a special-habitat plant species with strong resistance to stress, in this study, we speculated that the *C. rosea* Ca^2+^ signal system, especially CaM and CMLs, might play a crucial role in allowing this species to tolerate the extreme adversity of tropical and subtropical coastal regions, and the total *CrCaM*/*CrCML* family was retrieved based on the entire *C. rosea* genome sequencing data. In turn, the *CrCaM*/*CrCML* genes’ chromosomal locations and synteny and the proteins’ phylogenetic relationships were also summarized. In addition, the expression levels of *CrCaM*/*CrCML* were also analyzed using RNA-Seq analysis combined with quantitative reverse transcription PCR (qRT-PCR) detection. We cloned several *CrCaM*/*CrCML* genes and performed preliminary functional verification with a yeast heterogeneous expression system. This study provides the theoretical molecular mechanisms underlying *C. rosea*’s stress resistance, particularly mediated by the pivotal regulatory factors in the Ca^2+^ signal transduction pathway, the CaM/CML family. Our study also explores new possibilities and demonstrates experimental evidence for the molecular regulation and biological relevance of *CrCaM*/*CrCML*s in the adaptation of *C. rosea* to tropical and subtropical coastal habitats.

## 2. Results

### 2.1. Genome-Wide Identification and Annotation of the CaMs and CMLs in C. rosea

In total, seven CrCaMs and 44 CrCMLs were identified through Pfam, and SMART search predicted the presence of the EF-hand domain pair (PF13499). Their corresponding genes were finally selected in the order of their chromosomal locations (*CrCaM1* to *CrCaM7* and *CrCML1* to *CrCML44*) (Table 1). The amino acid length of the CrCaMs/CrCMLs proteins ranged from 81 aa (CrCML3) to 288 aa (CrCML6), the molecular weights ranged from 9.21 kDa (CrCML3) to 33.26 kDa (CrCML6), and the theoretical isoelectric points (PI) ranged from 3.89 (CrCaM3) to 8.79 (CrCML35) (Table 1). In addition, only CrCML9 (8.70), CrCML35 (8.79), and CrCML43 (7.63) were considered to be basic (PI > 7). The instability index (II) ranged between 59.01 (CrCML26) and 14.27 (CrCML44) and averaged approximately 35, and a small portion of members (19 members, 36%) had a high instability index (> 40). This result indicates that most of these members might be stable. The aliphatic index (AI) assessment shows that most members had lower values, indicating that only a small number of these proteins appeared to be lipophilic or hydrophobic. Accordingly, only one member (CrCML31) had calculated grand average of hydropathy (GRAVY) values greater than 0 (0.057), implying that most of these proteins were quite hydrophilic. The contents of the disordered amino acids reflected the regularity of protein structure, and we also predicted the 3D structures (Appendix A) and calculated the disordered amino acid contents of all of the CrCaM/CrCMLs. Obviously, seven CrCaMs presented typical dumbbell structures, and only half of the CrCMLs also showed dumbbell structures, while some of the CrCMLs presented disordered 3D structures, such as CrCML2, CrCML4, CrCML6, CrCML9, CrCML10, CrCML11, CrCML15, CrCML19, CrCML20, CrCML21, CrCML24, CrCML25, CrCML27, CrCML32, CrCML36, CrCML37, CrCML38, and CrCML43 (Appendix A). Compared with some typical intrinsically disordered protein family members identified from *C. rosea* [32,33], the disordered amino acid contents of the CrCaMs/CrCMLs were relatively low (most of them were below 50, Table 1), which indicated that the CrCaM/CrCMLs had stable 3D structures for their functional guarantee. Moreover, the subcellular localization prediction that the CrCaM/CrCML members were widely distributed within most of the cell organelles further supported their biological functional diversities and universalities.

### 2.2. The Genes’ Localization and Analysis of the CrCaM/CrCML Genes Structure

Because the *CrCaM*/*CrCML* genes were named based on their chromosomal locations, the seven *CrCaM*s and forty-four *CrCML*s were distributed on the eleven *C. rosea* chromosomes orderly (Figure 1). Overall, three *CrCaM*s (*CrCaM2*, *CrCaM3*, and *CrCaM4*) were located on chromosome 04, and *CrCaM1*, *CrCaM5*, *CrCaM6*, and *CrCaM7* were located on chromosomes 03, 06, 07, and 11, respectively. Chromosomes 01, 03, and 05 held more *CrCML*s (more than or equal to five members), chromosomes 02, 04, and 10 each contained four *CrCML* genes, and chromosomes 09 and 11 each contained two *CrCML* genes. Chromosome 08 contained only one *CrCML* gene (*CrCML36*).

Many plant genomes undergo duplication and expansion events that would, in turn, lead to an increase in the number of functional response genes and the overall size of some specific families or generate multiple paralogs of some genes, resulting in multiple versions of similar proteins within an organism [34]. The gene duplication events also could be considered to be important mechanisms for plant adaptive evolution at the genome level [35]. In this study, there were only two tandem duplication events (*CrCaM2*/*CrCaM3*, *CrCML43/CrCML44*) in the *CrCaM/CrCML* superfamily and fourteen segmental duplication events in *CrCML* subfamily (Table 2, Figure 2). The selection pressure imposed on the *CrCaM/CrCML* genes was assessed by calculating the ratio of non-synonymous (Ka) to synonymous (Ks) substitution values. We found that some *CrCML* genes were under evolutionary pressure, with a Ka/Ks ratio ranging from 0.0111528 (*CrCaM5*/*CrCaM7*) to 0.370378 (*CrCML14*/*CrCML21*) (Table 2). The Ka/Ks values of the gene pairs were all considerably lower than 1.0, which suggests that these gene pairs were primarily selected for purification during the evolutionary process with limited functional divergence after duplication.

Furthermore, we analyzed the gene structures of the *CrCaM*/*CrCML* genes by comparing their coding sequences with their corresponding genomic DNA sequences (Figure 3). Generally speaking, the *CrCaM*s had more complicated gene structures than the *CrCML*s. Seven *CrCaM*s all had from one to three introns, while only *CrCML6*, *CrCML17*, *CrCML18*, *CrCML19*, *CrCML23*, *CrCML26*, and *CrCML43* had introns. Among them, *CrCML26* and *CrCML43* contained only one intron, *CrCML17*, *CrCML18*, *CrCML19*, and *CrCML23* contained three introns, and *CrCML6* contained five introns. Most of the *CrCML*s contained no introns. Obviously, the gene structure of the *CrCML*s was also relatively simple, which indicates that their transcription and subsequent translation might be rapidly activated under different stress challenges because of the formation of mature mRNA of the *CrCML*s reduces the steps for the elimination of introns, thus shortening the response time. This further fortifies the necessary functions of *CrCML*s in environmental stress responses.

### 2.3. The Conserved Motifs of Proteins and Phylogenetic Analysis

The CrCaM/CrCML conserved motifs were also demonstrated using MEME analysis, and ten distinct motifs were identified (Figure 4). Motif 1 represents the typical EF-hand, and it was present in all CrCaMs with four repetitions. Most of the CrCMLs (23 of 44 CrCMLs) also contained four EF-hand motifs, while their conservativeness was inferior to that in the CrCaMs. In some CrCMLs (CrCML4, CrCML8, CrCML16, CrCML28, CrCML30, CrCML37, CrCML40, CrCML41, and CrCML42), motifs 2, 4, 7, and 10 replaced the EF-hand motif, which might indicate that these CrCMLs are more diverse than the CaM proteins. Therefore, the complicated and changeable functions of the CrCML members can also be further inferred.

We first aligned the CrCaM/CrCMLs to each other and performed their phylogenetic analysis (Figure 5). Undoubtedly, the seven CrCaMs were clustered into a small subgroup with a close evolutionary trend as CrCML16, CrCML17, CrCML18, CrCML23, and CrCML24. The CrCML10 seemed to be unique, and CrCML3, CrCML6, CrCML19, and CrCML35 also formed a relatively independent cluster compared to other CrCaM/CrCMLs. Overall, the CrCaM proteins were highly conserved, and half of the CrCMLs had high levels of similarity within protein pairs (over 90%, eleven pairs) that were probably generated with gene duplication.

To characterize the phylogenetic relationships of all of the CrCaM/CrCMLs with similar members in the other species, we also inferred the phylogenetic relationships of the CaM/CMLs among *C. rosea*, rice, and Arabidopsis using the protein sequences of 57 AtCaM/AtCMLs, 37 OsCaM/OsCMLs, and 51 CrCaM/CrCMLs into seven cluster groups (containing 49, 23, 14, 37, 13, 2, and 5 members in each group) (Figure 6). These groups were designated as I to VII, containing 13, 11, 4, 15, 5, 1, and 2 CrCaM/CrCMLs, respectively. Overall, the CrCaMs showed relatively high levels of similarity with those CaMs of other species, and they were highly conserved. The CMLs demonstrated relatively higher variability and diversity than CaMs. This was also consistent with the conserved motif analysis using MEME (Figure 4).

### 2.4. Abiotic Stress-Related Cis-Regulatory Elements (CEs) in the CrCaM/CrCML Promoters

To gain further insights into the regulatory mechanisms of *CrCaMs*/*CrCMLs* responding to environmental stresses and developmental signals, the promoter features of the *CrCaMs*/*CrCMLs* were systematically identified with different CE categories. Phytohormone and stress responses *cis*-regulatory elements in the promoter regions of the 51 *CrCaMs*/*CrCMLs* genes were predicted (Figure 7).

The results reveal that the CEs associated with phytohormone responses, including abscisic acid (ABREs), auxin (auxin-responsive elements), gibberellin (gibberellin-responsive elements), methyl jasmonate (MeJA-responsive elements), salicylic acid (salicylic acid-responsive elements), and ethylene (EREs), existed extensively in most of the *CrCaM*s/*CrCML* promoter regions. Some adversity-related CEs, such as anaerobic-responsive elements, low temperature, heat stress elements (HSE), and defense and stress-related elements (TC-rich repeats), also occurred widely in the promoter regions of the *CrCaM*s/*CrCML*s. The *CrCaM*s and *CrCML*s had the same or different CEs, indicating that these genes may be regulated in response to stress, sometimes simultaneously, or that these genes especially respond to adverse external environments.

### 2.5. Tissue- and Habitat-Specific Expression Profiles of the CrCaMs/CrCMLs

The expression patterns of the *CrCaMs*/*CrCMLs* were detected first with an RNA sequencing analysis concerned with tissue-specific patterns, and the purpose of this was to confirm the real transcripts of the *CrCaMs*/*CrCMLs* that might be involved in *C*. *rosea* growth and development. The expression levels of the *CrCaMs*/*CrCMLs* in five different tissues (including the roots, vines, leaves, flower buds, and young fruit) were calculated and displayed using a heatmap (Figure 8A) according to the RNA-Seq data (Appendix A). Nearly one-third of the *CrCaMs*/*CrCMLs* showed relatively higher expression levels in both roots and flower buds compared to that in vines, leaves, and young fruit, with log2-transformed FPKM values (FPKM+1) greater than 4.00 (Figure 8A). In young vine tissues of *C. rosea* plants, only *CrCaM2*, *CrCaM5*, *CrCaM6*, *CrCaM7*, *CrCML4*, *CrCML10*, *CrCML16*, *CrCML28*, *CrCML29*, and *CrCML41* presented relatively high expression levels, while in leaf tissues, only *CrCaM2*, *CrCaM5*, *CrCaM6*, *CrCaM7*, *CrCML1*, *CrCML4*, *CrCML10*, *CrCML16*, *CrCML24*, *CrCML28*, *CrCML41*, and *CrCML44* were relatively highly expressed. In young fruit tissues of adult *C. rosea*, the global expression levels of most *CrCaMs*/*CrCMLs* showed relatively lower levels, and only *CrCaM5*, *CrCaM6*, *CrCaM7*, *CrCML16*, *CrCML28*, *CrCML29*, *CrCML34*, and *CrCML41* were expressed in the fruit at high levels, indicating that these two genes may be involved in the reproduction of *C*. *rosea* plants.

*C. rosea* is a typical tropical leguminous halophyte, and this species shows great growth advantages on tropical coral islands and reefs with excellent tolerance to heat, drought, high salinity/alkalinity, strong sunlight, and high ultraviolet radiation. Halophytes are plants that can complete their life cycles under very high salt concentrations and even benefit from this special habitat [36]. It has been reported that the expression of halophyte-derived genes improves the tolerance of transgenic plants in response to various stresses [37,38]. In this regard, we performed a gene expression analysis of two mature leaf samples gathered from SCNBG (optimal habitat) and YX Island (special habitat) (Figure 8B). Except for some of the *CrCML* genes, which showed extremely low or even undetectable expression levels in mature leaves of both samples, including *CrCML3*, *CrCML7*, *CrCML12*, *CrCML14*, *CrCML15*, *CrCML23*, *CrCML25*, *CrCML26*, *CrCML29*, *CrCML31*, *CrCML34*, *CrCML35*, *CrCML39*, *CrCML42*, and *CrCML44*, the others all showed varying transcripts both in the SCNBG and the YX tissue samples. More obviously, several genes showed relatively higher expression levels in the YX sample, including *CrCML1*, *CrCML10*, *CrCML16*, *CrCML24*, *CrCML27*, *CrCML32*, *CrCML33*, and *CrCML43*, and only *CrCaM2*, *CrCaM3*, *CrCML5*, *CrCML11*, *CrCM17*, *CrCML18*, *CrCML21*, *CrCML37*, and *CrCML40* presented slightly lower expression levels in the YX sample than that in the SCNBG sample. This result indicates that some *CrCaM*s/*CrCML*s might play positive protective roles in *C*. *rosea*’s adaptation to coral reef habitats. The RNA-sequence data of the *CrCaM*s/*CrCML*s are listed in Appendix A.

### 2.6. Expression Profile of the CrCaM/CrCML Genes in Response to Abiotic Stress

A promoter analysis of the *CrCaM*/*CrCML* genes indicated that the transcriptional regulatory model of this gene family was diverse and member-specific. Plant *CaMs*/*CMLs* have been proven to be modulated by multiple environmental stressors, including heat, cold, salt, drought, or heavy metals [1,18,19], and some individual plant *CaM* or *CML* genes have been proven to mediate stress tolerances by modulating downstream co-factors [20,21,22,23,24,25,26].

The *CaM* or *CML* genes isolated from special-habitat plants that had heterologous transgenic experiments performed provided further evidence of their clear functions for environmental stress responses [24,27]. *C. rosea* is a unique plant that is adapted to specialized habitats, thriving especially on coral reefs. This suggests that the species can withstand various environmental challenges, such as drought, high osmotic pressure, salinity/alkalinity, heat, and intense bright light or high ultraviolet radiation. Furthermore, whether the halophyte *C. rosea*’s *CaM*/*CML* genes can also function in multiple environmental conditions has also become an interesting research topic. To recognize the role of heat-related signaling responses in the regulation of *CrCaM*s/*CrCML*s, an expression analysis was performed with plants treated with heat (42 °C) using RNA-sequencing. Both in the root and in leaf samples, more than half of the *CrCaM*s/*CrCML*s exhibited obvious expression alterations and were up- or down-regulated by heat challenge (Figure 9). In general, in both the root and leaf samples, a large proportion of the *CrCaM*s/*CrCML*s showed decreased expression levels under heat stress. However, in two *C. rosea*’s tissues, several noticeably up-regulated *CrCaM*s/*CrCML*s members raised concern, including *CrCaM2*, *CrCML7*, *CrCML32*, and *CrCML43* in roots and *CrCML4*, *CrCML8*, *CrCML22*, *CrCML24*, *CrCML27*, *CrCML32*, and *CrCML43* in leaves. This suggests that the above *CrCaM*/*CrCML* members were genes with different but unique potential biological functions in the heat responses of *C. rosea* plants.

We further analyzed the expression of all *CrCaM*/*CrCML* family members in *C. rosea* seedling plants subjected to high salinity, alkaline, and high osmotic (mimic drought) stresses. Overall, we noticed that the modulation of specific *CrCaM*s/*CrCML*s showed completely different expression patterns, either in roots (Figure 10A) or in leaves (Figure 10B). We also reconfirmed the expression patterns of some candidate *CrCaM*s/*CrCML*s with qRT-PCR, and the results also demonstrated that heat, high salinity, alkaline, osmotic stress, and oxidative stress could rapidly induce the transcript changes in the roots, vines, and leaves of *C. rosea* plants (Figure 11). In summary, under heat stress, the *CrCaM*s/*CrCML*s exhibited more pronounced expression changes in the roots compared to the vines and leaves. However, in response to other abiotic stresses, such as high salinity/alkalinity, high osmotic pressure, and oxidative stress, a larger number of *CrCaM*s/*CrCML*s showed greater expression changes in the leaves, particularly after prolonged stress exposure (L-48 h, 2d), compared to the roots (Figure 10 and Figure 11). Notably, several members, including *CrCaM2*, *CrCaM3*, and *CrCaM4*, exhibited substantial expression changes under most stress challenges, making them particularly intriguing for further study. Further identification of stress-regulated expressed patterns in the *CrCaM*s/*CrCML*s can provide more information on the ecological suitability for extreme adversity of *C. rosea* plants.

### 2.7. Functional Characterization of the CrCaMs/CrCMLs in Yeast

To provide relevant biological significance combined with the in silico expression meta-analysis of the *CrCaMs*/*CrCML*s, the initial functional identification of some *C. rosea CaM*/*CML* genes was investigated using the yeast heterogeneous expression system. In short, the cDNAs’ open reading frames of *CrCaM1*, *CrCaM3*, *CrCaM6*, and *CrCaM7* and *CrCML10*, *CrCML16*, *CrCML24*, *CrCML27*, *CrCML32*, and *CrCML33* were PCR cloned and inserted into the expression cassette of pYES2 under a galactose-induced promoter. The WT yeast was then transformed with recombinant CrCaMs/CrCMLs-pYES2 vectors, and stress tolerance tests were performed that included heat (52 °C), high salinity (NaCl), alkaline (NaHCO_3_), high osmotic challenge (sorbitol), and oxidative stress (H_2_O_2_).

To explore the role of *C. rosea CaM*/*CML* in the regulation of stress tolerance in vivo, the growth of yeast strains containing recombinant CrCaMs/CrCMLs-pYES2 vectors was first compared with control cells (transformed with empty vector pYES2) after a high-temperature challenge (52 °C). The H_2_O_2_-sensitive mutant strain, *skn7∆,* is also sensitive to heat, and the 52 °C treatment for 15 min was nearly lethal for *skn7∆*, while all the *CrCaM/CrCML* members improved the survival rate of the *skn7∆* strain to varying degrees. In addition, under prolonged heat (52 °C for 20 min), some of the *CrCaM/CrCML* genes even improved the growth status of *skn7∆* better than the WT (Figure 12A). From this, we can conclude that in the single-cell yeast system, the heat damage was mitigated to different extents by an accumulation of CrCaM/CrCMLs. Unexpectedly, the results indicate that the CrCaM/CrCMLs did not seem to elevate the H_2_O_2_-tolerance of *skn7∆* (Figure 12B), and this might imply another antioxidant signaling pathway by the calcium signals.

The plants’ *CaM*s and *CML*s have been shown to be involved in diverse cellular processes, including signaling and other different abiotic stress responses [10]. We also detected the salt (Figure 12C), alkaline (Figure 12D), and high osmotic stress (Figure 12E) tolerance of WT yeast. The tested *CrCaM/CrCML* genes all improved the NaCl tolerance of yeast to different degrees, and *CrCaM3*, *CrCaM6*, and *CrCML27* appeared to possess the best tolerability (Figure 12C). However, for alkalinity, it seemed that *CrCaM6* and *CrCML16* did not affect the alkali tolerance of yeast, while other genes all improved the alkali tolerance to varying degrees (Figure 12D). As for high osmotic stress, it seemed that only *CrCML10* and *CrCML27* slightly elevated the osmotic tolerance of yeast (Figure 12E). The above experiments were repeated at least three times.

## 3. Discussion

Calcium (Ca^2+^), a universal second messenger ion, plays a key role in the signal transduction process during plant growth, development, and the stress response [39]. Plants have evolved a series of physiological and biochemical mechanisms to cope with environmental and developmental stimuli. One of the most important ways uses the Ca^2+^ recruited by proteins’ EF-hand domains, and then the protein-Ca^2+^ complex acts as a messenger in response to a given stimulus [40]. Accordingly, the Ca^2+^-binding proteins, such as CaMs and CMLs, which contain conserved (CaMs) or degenerated (CMLs) EF-hand domains, might be key integrators for the Ca^2+^ signal pathway, thereby regulating plants’ development or response to internal or environmental stimuli. Thus, it is necessary to explore the pathways or mechanisms of plant CaMs/CMLs responding to developmental signals or stresses. The special habitat plant species, *C. rosea*, might be a potential research model for investigating the mechanisms for plant tolerance to extreme stresses. Previously, we identified a series of *C. rosea*’s gene families and primarily focused on the water shortage stress [16,32], heat shock response [41,42], and heavy metal detoxification of this species as a halophyte [33,43,44,45]. In this study, to address the possible roles of the *C. rosea* Ca^2+^ signal system for its ecological adaptability to tropical coral extreme habitats, we retrieved the *CrCaM*/*CrCML* gene family from the entire *C. rosea* genome. In brief, the *C. rosea* proteome was searched with the conserved EF-hand domain (PF13499), and the *CrCaM/CrCML* gene family was identified. The related evolutionary relationships, sequence features, and duplication events were also explored systematically.

Until now, at least four leguminous plants’ *CaM* or *CML* gene families have been systematically identified at the genome-wide level, and these include *Lotus japonicus* (7 *LjCaM*s and 19 *LjCML*s) [46], *Medicago truncatula* (50 *MtCML*s) [47], *Glycine max* (soybean) (41 *GmCML*s) [48], and *Phaseolus vulgaris* (common bean) (111 *PvCML*s) [49]. These genes are all supposed to be involved in plant development according to RNA-Seq assays with different tissues or organs, and some of them might also mediate abiotic or biotic stress responses [46,47]. On the whole, the related research demonstrated that the number of *CaM*s in plants is basically below ten, while the number of *CML*s in plants is potentially dozens, even up to one hundred [46,47,48,49]. A previous report indicated that the varied gene number of plants’ *CaM* or *CMLs* and in genomes might be dependent on evolutionary pressure, functional requirements, and the complexities of the plant [9]. This, combined with the gene structures of *CrCaM*s or *CrCML*s (containing more introns in *CaM*s while fewer or no introns in *CML*s) (Figure 3), allows us to propose that *CrCaM*s and *CrCML*s might play different cellular and ecological roles associated with *C. rose*’s adaptation and withstand the different evolutionary pressures of this species to its native habitat.

Some specific *CaM* or *CML* gene members have been systematically investigated for biological functions via transgenic assays. For example, in the leguminous alfalfa plant, the overexpression of *Medicago sativa MsCML46* enhances transgenic tobacco’s tolerance to freezing, drought, and salt stresses by improving the contents of osmotic regulatory solutes and antioxidant enzyme activity while decreasing reactive oxygen species (ROS) accumulations [22]. Another alfalfa (*Medicago truncatula*) *CML* gene, *MtCML42*, can prompt early flowering by up-regulating the flowering-time gene, *MtFTa1*, and mediating cold tolerance by affecting stress-resistant gene expressions such as *MtABI5*, *MtCBF1*, and *MtCBF4* [50]. In crops, some *CaM* or *CML* genes have also been proven to possess positive regulating effects for plant resistance. The *Cucumis melo CmCML13* can enhance the salt tolerance of transgenic Arabidopsis through reducing the shoot Na^+^ content and also improving the drought resistance [21]. The ectopic expression of the finger millet (*Eleusine coracana*) CaM gene, *EcCaM*, confers improved drought and salinity tolerance in Arabidopsis by regulating ion leakage, the proline content, and the antioxidant capacity [23]. The potato (*Solanum tuberosum*) CaM gene, *StCaM2*, was induced by multiple abiotic stresses in potato tissues, and its overexpression in transgenic tobacco plants can enhance their tolerance to salinity and drought stress [26]. Conversely, some CaM genes demonstrated the opposite effect for stress resistance; for example, the barley (*Hordeum vulgare*) CaM gene, *HvCaM1*, negatively regulates salt tolerance. Although its expression is significantly induced by long-term salt stress in barley tissues, the overexpression (OE) of *HvCaM1* in barley resulted in the reduced salt tolerance than *HvCaM1* knockdown (RNA interference) lines and the WT [20]. From this, we can conclude that, for the specific *CaM* or *CML* genes, their biological functions might be unique, and even their expression is obviously induced under stress challenges. Thus, the stress tolerance caused by the overexpression of these genes in target plants may exceed expectations. Because their biological functions could be contradictory, further research regarding unknown plant *CaM* or *CML* genes may prove to be challenging.

*C. rosea* is a highly resistant plant species distributed primarily in tropical or subtropical coastal saline regions [15]. Due to its strong vitality and enormous ecological landscape benefits for islands and reefs, this species has caused extensive concern in recent years. In this research, from the point of the phylogenetic analyses of *CrCaM*s and *CrCML*s, the protein structure prediction, the gene expression summary, and some transgenic assays with yeast, we expounded comparative gene families and discussed the function of these genes for the ecological adaptation of *C. rosea*, as well as exploring and evaluating the possibilities for future crop genetic improvements with these genes. In fact, there has been some exploratory research regarding the biological functions of *CaM* or *CML* genes from special habitat plants. The semi-mangrove *Millettia pinnata* showed excellent adaptation to the saline environment, and the salt-responsive CML gene, *MpCML40,* was induced by salt treatment in the *M. pinnata* plant. The *MpCML40* over-expressed transgenic Arabidopsis presented highly enhanced the seed germination rate and root length under salt and osmotic stresses, further clarifying the positive regulating roles of *MpCML40* in response to salt stress and potential application values for generating salt-tolerant crops [24]. *Paspalum vaginatum* O. Swartz is a typical seashore paspalum halophyte in tropical and subtropical coastal regions worldwide, and this species has exuberant vitality even under persistent high-salinity/alkaline stresses. A *CML* gene from *P. vaginatum*, *PvCML9*, was highly expressed in *P. vaginatum* roots and stems, and over-expression of *PvCML9* in rice and paspalum led to reduced salt tolerance. Instead, down-regulating the expression of this gene in paspalum showed increased salt tolerance [27]. *Nitraria sibirica* Pall. (*N. sibirica*) is a typical halophyte with strong adaptability to extreme saline-alkali and drought environments. Some of the *N. sibirica CaM*/*CML* genes showed considerably regulated expression patterns due to high salt challenges, and the overexpression of two *NsCML*s was found to mitigate H_2_O_2_ accumulation caused by abiotic stresses, thereby regulating ROS homeostasis and further indicating the possible molecular mechanisms of *NsCML*s-mediated abiotic stress tolerance in *N. sibirica* [31]. A precious woody species, *Phoebe bournei*, is a subtropical evergreen broad-leaved tree species. Currently, this species is faced with seriously threats to its survival due to habitat degradation, primarily due to contentious drought challenges. The *PbCaM*s/*PbCML*s might be related to the stress responses of *P. bournei*, and the overexpression of *PbCaM3*/*PbCML13* genes significantly increased the tolerance of yeast cells to drought stress [51].

Although the *CrCaM* and *CrCML* genes were evolutionary conserved, their biological functions might be unique. Heat, salt, high alkalinity, and drought are also major environmental challenges that limit plant growth and development in the special habitats of tropical islands or reefs, and we propose that *CrCaM* and *CrCML* genes might be recruited to cope with multiple stress challenges in the special habitats of *C. rosea* plants. In this study, we also cloned several *CrCaM* and *CrCML* genes and ectopically expressed them in yeast for functional confirmation. Opposite to our expectation for heat tolerance, all of the detected *CrCaM*s and *CrCML*s improved the heat tolerance of the H_2_O_2_-sensitive mutant yeast strain *skn7∆*, while the H_2_O_2_ tolerance of *skn7∆* was nearly unaffected (Figure 11A and 11B). Similarly, the high salinity and alkaline tolerance of the WT yeast strain was also changed (up- or down- regulated) by the expression of *CrCaM*s and *CrCML*s (Figure 11C and 11D), while the high osmotic stress tolerance remained largely unchanged ((Figure 11E). This is quite different from other *C. rosea* molecular chaperone genes, such as *CrLEA*s/*ASR*s [32], or *CrCYSTM*s [33]. From this, we can also conclude that the functions of specific plant *CaM* or *CML* genes are unique and require further individual investigations.

In summary, this study identified the *C. rosea CrCaM* and *CrCML* gene family, consisting of seven novel *CaM* members and 44 *CML* members. Sequence analysis showed that of all CrCaM and CrCMLs contained the EF-hand domain (PF13499). We also conducted RNA-Seq analysis to examine the gene and promoter sequences of the *CrCaM* and *CrCML* family, aiming to understand their biological functions under abiotic stresses, as well as the natural ecological adaptability of *C. rosea* plants to tropical and subtropical coastal areas. Overexpression of certain *CrCaM* and *CrCML* genes in yeast resulted in altered tolerance to heat, salt, and high alkalinity stress. These findings offer comprehensive insights into the *C. rosea CrCaM* and *CrCML* gene family, paving the way for further functional studies on plant CaMs/CMLs. Additionally, these results will support further research on the role of CrCaMs/CrCMLs as protective molecules and mediators of responses to environmental conditions and to internal developmental signals.

## 4. Materials and Methods

### 4.1. Plant Materials and Stress Treatments

The *C. rosea* seedlings were cultivated with seeds gathered from the coastal regions of Hainan Province, China. The plant growth conditions were maintained in an artificial greenhouse system (22 °C with a photoperiod of 16 h light/8 h darkness) in the South China National Botanical Garden (SCNBG, 23°18′ N, 113°37′ E). The tissues of *C. rosea* plants were also captured from Yongxing Island (YX, 16°83′ N, 112°34′ E) and the South China National Botanical Garden (SCNBG) for gene expression detection. One-month-old *C. rosea* seedlings were used for the different abiotic stress treatments. In brief, the seedling roots were soaked in 600 mM NaCl, 150 mM NaHCO_3_ (pH 8.2), 300 mM mannitol, a 45 °C pre-warmed 1/2 Hoagland solution, and 10 mM H_2_O_2_ for high salinity, alkaline, high osmotic, heat, and oxidative stress treatments, respectively. Plant tissues were collected at different time points (2 h and 48 h/2 d for the RNA-Seq or qRT-PCR). Three independent biological replicates were used.

### 4.2. Identification and Evolutionary Analyses of the CrCaM/CrCML Family in C. rosea

All of the collected genome sequencing data were submitted to GenBank (Accession No.: JACXSB000000000). The Pfam ID (EF-hand domain pair, PF13499) was used to search for the CrCaM/CrCML family members, and putative sequences of the CrCaM/CrCMLs were identified and submitted to the National Center of Biotechnology Information (NCBI) Conserved Domain Database (https://www.ncbi.nlm.nih.gov/Structure/cdd/wrpsb.cgi, accessed on 10 April 2024) to confirm the presence of the EF-hand domain. The sequence information is summarized in Appendix A.

The amino acid sequences of CrCaMs/CrCMLs were aligned using ClustalW software by MEGA-X software, and a phylogenetic tree was constructed using the neighbor-joining (NJ) phylogenetic method with 1000 bootstrap replicates. The CrCaMs/CrCMLs were also compared with those from other plant species using a multiple sequence alignment conducted using ClustalW software and constructed the phylogenetic tree with MEGA-X. The CaM/CML sequences of Arabidopsis (7 AtCaMs and 50 AtCMLs) [12] and rice (5 OsCaMs and 32 OsCMLs) [13] were downloaded from TAIR and RAP-DB (https://rapdb.dna.affrc.go.jp/, accessed on 10 April 2024), respectively. The amino acid sequences of these three species’ CaMs/CMLs were used to construct the phylogenetic tree.

### 4.3. Chromosomal Location, Ka/Ks Calculation of the CrCaM/CrCML Genes, and Conserved Motifs of the Proteins

All of the *CrCaM*/*CrCML* family members were mapped to the chromosomes and scaffolds of *C. rosea* based on their physical location information. The localization of genes was visualized through MG2C 2.1 (http://mg2c.iask.in/mg2c_v2.1/, accessed on 10 April 2024) and Adobe Illustrator. The synonymous and non-synonymous substitution rates (Ks and Ka, respectively) and the probability (*p*-value) of Fisher’s exact test of neutrality were calculated, and the selective pressures on the duplication of *CrCaM*s/*CrCML*s based on all nucleotide sequences were explored using the Nei-Gojobori model with 1000 bootstrap replicates [52]. The gene duplication events were analyzed using the Multiple Collinearity Scan toolkit (MCScanX, http://chibba.pgml.uga.edu/mcscan2/, accessed on 10 April 2024) with default parameters, and tandem duplications were identified manually (Table 2). The gene structures, including the intron-exon patterns and the 5’ and 3’ untranslated region (UTR) of the *CrCaM*/*CrCML* genes, were displayed through the GSDS2.0 server (http://gsds.gao-lab.org, accessed on 10 April 2024).

To further investigate the features of the CrCaMs/CrCMLs, conserved motifs were analyzed using the MEME program (Multiple Em for Motif Elicitation, http://meme-suite.org/index.html, accessed on 10 April 2024) with the following parameters: motif number was set to 10, classic motif discovery mode, and any number of repetitions.

### 4.4. Cis-Regulatory Element Analysis of the CrCaM/CrCML Promoters

The predicted *cis*-regulatory elements (CEs) were scanned using the PlantCARE program (http://bioinformatics.psb.ugent.be/webtools/plantcare/html/, accessed on 10 April 2024) by searching for the promoter regions (2000 bp upstream from the translation start site ATG) of all the *CrCaM*s/*CrCML*s. These CEs were classified into hormone-specific (ABREs, auxin-responsive elements, gibberellin-responsive elements, MeJA-responsive elements, salicylic acid-responsive elements, and EREs) and environmental or development-responsive (anaerobic-responsive elements, as-1, HSE, LTRE, TC-rich repeats, MYB, and MYC), and are summarized in Appendix A. The CE analysis results were visualized and mapped to the *CrCaM*/*CrCML* promoters using TBtools software (version: 0.67) [53].

### 4.5. RNA-Seq of the CrCaMs/CrCMLs in Different C. rosea Tissues or Under Different Stress Treatments

Tissue-specific expression profile analyses during different developmental stages of the *C. rosea CrCaM*s/*CrCML*s were conducted using Illumina HiSeq X sequencing technology. Five different tissues from *C. rosea* plants (root, vine, young leaf, flower bud, and young silique samples) were collected from *C. rosea* adult plants and seedlings growing in the SCNBG. Mature leaf samples from *C. rosea* growing in SCNBG and on YX Island were examined using FastQC (http://www.bioinformatics.babraham.ac.uk/projects/fastqc/, accessed on 10 April 2024) based on the primary 40 Gb clean reads and were mapped to the *C. rosea* reference genome using Tophat v.2.0.10 (http://tophat.cbcb.umd.edu/, accessed on 10 April 2024). For the expression profiles of the *CrCaM*s/*CrCML*s under different abiotic stressors, *C. rosea* seedling tissues (including leaves and roots) were also collected and sequenced at the transcriptome level. All of the expressed sequence tag (EST) information was then mapped to the *C. rosea* reference genome. Gene expression levels were calculated as fragments per kilobase (kb) of transcript per million mapped reads (FPKM) according to the length of the gene and the read counts mapped to the gene: FPKM = total exon fragments/[mapped reads (millions) × exon length (kb)]. The expression levels of the *CrCaM*s/*CrCML*s were visualized as clustered heatmaps of log2 (FPKM + 1) values using TBtools. The FPKM values for all samples are listed in Appendix A.

### 4.6. Expression Pattern Analysis Using Quantitative Reverse Transcription (qRT)-PCR

The total RNA was extracted from *C. rosea* tissues for the qRT-PCR assays using the EasyPure® Plant RNA Kit (TransGen Biotech, Beijing, China), and the cDNA was then synthesized using cDNA Synthesis SuperMix (TransGen Biotech, Beijing, China) according to the manufacturer’s instructions. Quantitative RT-PCR was conducted using the LightCycler480 system (Roche, Basel, Switzerland) and TransStart Tip Green qPCR SuperMix (TransGen Biotech, Beijing, China). Reaction systems were prepared in 10 µL volumes as follows: 5 µL of 2 × Tip Green qPCR SYBR Master Mix, 0.5 µL of the forward primer (10 µM), 0.5 µL of the reverse primer (10 µM), 1 µL of cDNA (20 to 50 ng µL^−1^), and 3 µL of RNase-free water. Each sample was tested using three technical replicates to ensure the accuracy of the results. There were three biological replicates per treatment. The primers used for the qRT-PCR were designed using Primer 3.0 and are listed in Appendix A. The *CrEF-1α* gene was used as a reference, and the 2^−∆∆CT^ method was used to calculate the relative changes in gene expressions between the control and treatment plants [30].

### 4.7. Functional Identification Using a Yeast Expression System

The open reading frames (ORFs) of the *CrCaM*s/*CrCML*s were PCR amplified from different cDNA samples of *C. rosea* with gene-specific primer pairs (listed in Appendix A). After several PCR procedures, the PCR fragments were purified and cloned into the *Bam*HI and *Eco*RI sites of pYES2 to yield recombinant plasmids of CrCaMs/CrCMLs-pYES2 and sequenced. The *Saccharomyces cerevisiae* wild-type (WT) strain BY4741 (Y00000) and mutant strain *skn7Δ* (Y02900) were obtained from Euroscarf (London, UK) (http://www.euroscarf.de, accessed on 10 April 2024). The plasmids were introduced into yeast using the LiAc/PEG method. Yeast growth and stress tolerance tests were performed as previously described [16,32]. Single colonies of yeast transformants were selected and used to inoculate a liquid synthetic drop-out uracil medium with a 2% galactose (SDG-Ura) medium. It was then incubated overnight or longer at 30 °C, diluted with fresh pre-warmed SDG medium (volume ratio 1:10), and incubated with vigorous shaking for approximately 30 h at 30 °C to reach an optical density of one at OD600 (optical density at 600 nm). The cells were then serially diluted in 10-fold steps, and 2-μL aliquots of each was finally spotted onto the SDG medium plates with or without stressors. To determine the heat tolerance, the liquid yeast cultures were incubated in a constant temperature bath (52 °C) for different durations (*skn7Δ* for 15 and 20 min). The cultures were then spotted on solid SDG medium plates. The plates were incubated at 30 °C for two to five days and photographed.

### 4.8. Statistical Analyses

All analyses were conducted at least in triplicate, and the results are shown as the means ± SDs (n ≥ 3). The Excel 2010 (Microsoft Corporation, Albuquerque, NM, USA) statistics program was used to perform the statistical analyses.

## 5. Conclusions

In summary, this study identifies the *C. rosea CrCaM* and *CrCML* gene family, consisting of 7 novel CaM members and 44 CML members, mainly based on sequence analyses and RNA-Seq. This study further combined several experimental means, including qRT-PCR and overexpression assays in yeast, for initial function exploration of multiple specific *CrCaM* and *CrCML* members. The results show that the *C. rosea* CrCaMs/CrCMLs were evolutionarily distinct, and the CrCaMs were more conserved than the CrCMLs. The diverse expression of *CrCaM*s/*CrCML*s suggests that they might play specific roles in responding to the development of different *C. rosea* organs or various stress challenges. Overexpression of certain *CrCaM*s and *CrCML*s in yeast resulted in altered tolerance to heat, salt, and high alkalinity stress. These findings systematically offer comprehensive insights into the *C. rosea* CrCaM and CrCML gene family, and pave the way for further functional studies on plant CaMs/CMLs. Additionally, this study also primarily discusses the possible roles mediated by CrCaMs/CrCMLs for the natural ecological adaptability of *C. rosea* plants to tropical and subtropical coastal areas.

## Figures and Tables

**Figure 1 ijms-25-11725-f001:**
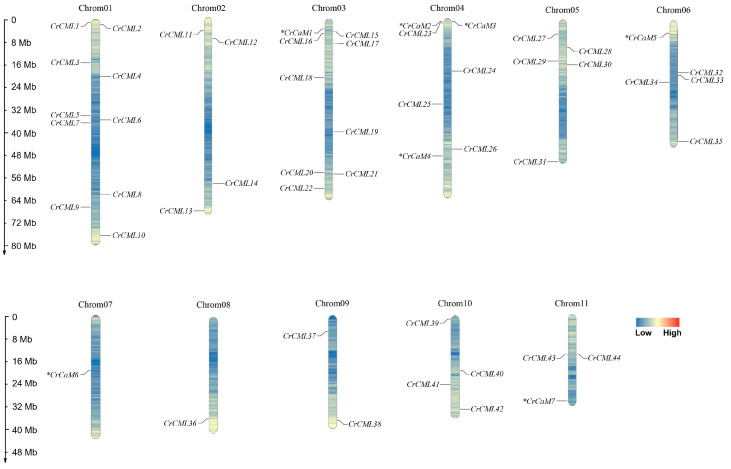
Locations of the 51 *CrCaM*s/*CrCML*s on 11 chromosomes of *Canavalia rosea*. Seven *CrCaM* genes were labeled with *.

**Figure 2 ijms-25-11725-f002:**
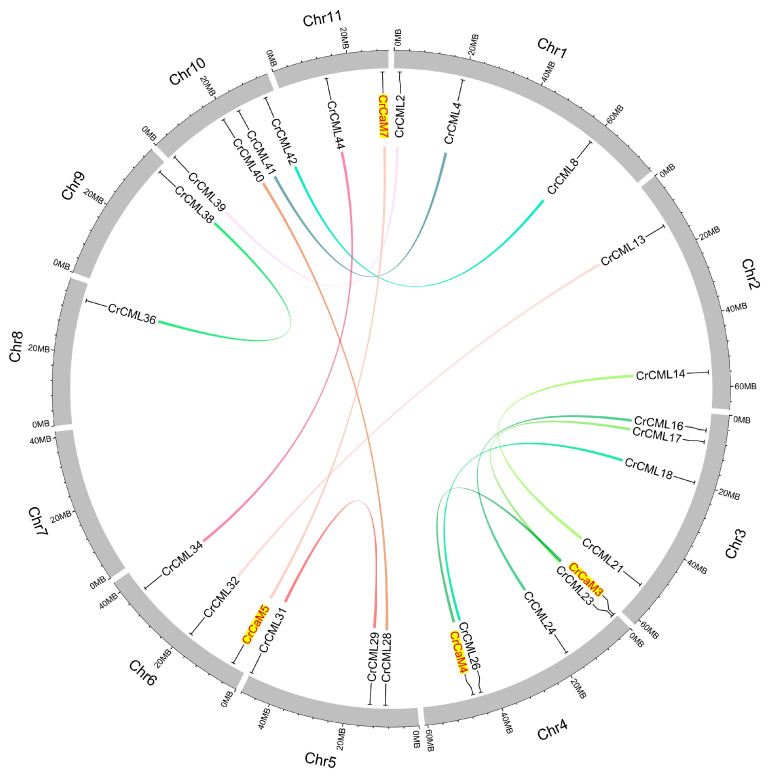
The distribution of segmental duplication of *CrCaM*s/*CrCML*s in *Canavalia rosea* chromosomes. Two *CrCaM* gene pairs (*CrCaM3* and *CrCaM4*, *CrCaM5* and *CrCaM7*) were marked with yellow background and red font.

**Figure 3 ijms-25-11725-f003:**
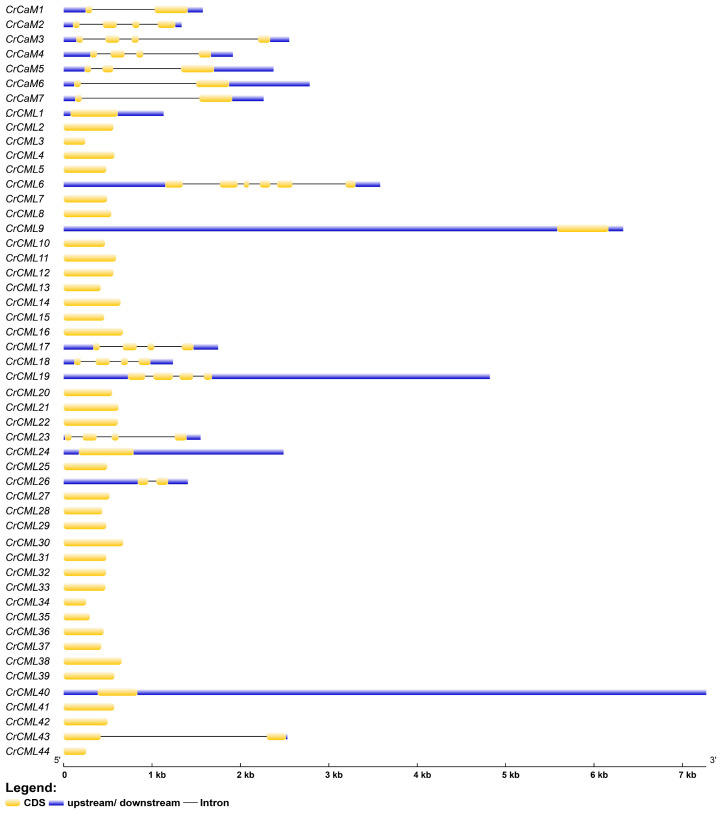
Exon–intron structures of the *CrCaM*s/*CrCML*s predicted using the Gene Structure Display Server (GSDS, http://gsds.cbi.pku.edu.cn/, accessed on 10 April 2024).

**Figure 4 ijms-25-11725-f004:**
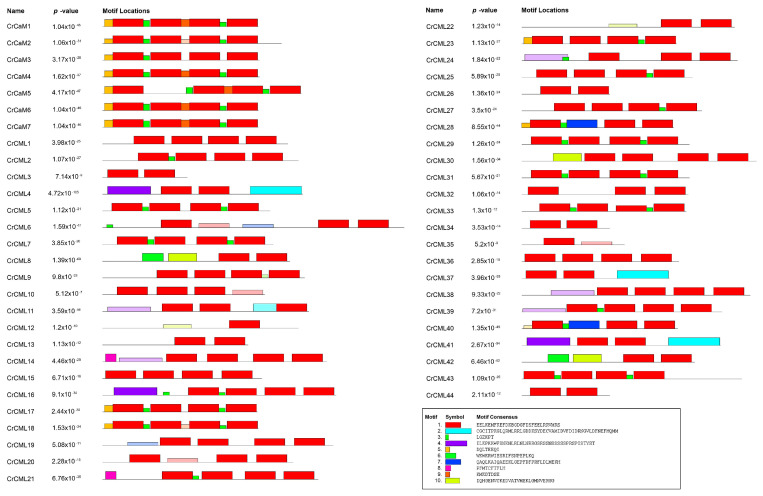
Structural analysis of the CrCaM/CrCMLs. The conserved motifs of each group identified using the MEME web server (http://meme-suite.org/index.html, accessed on 10 April 2024). Different motifs are represented by different colored boxes, and the motif sequences are provided at the bottom.

**Figure 5 ijms-25-11725-f005:**
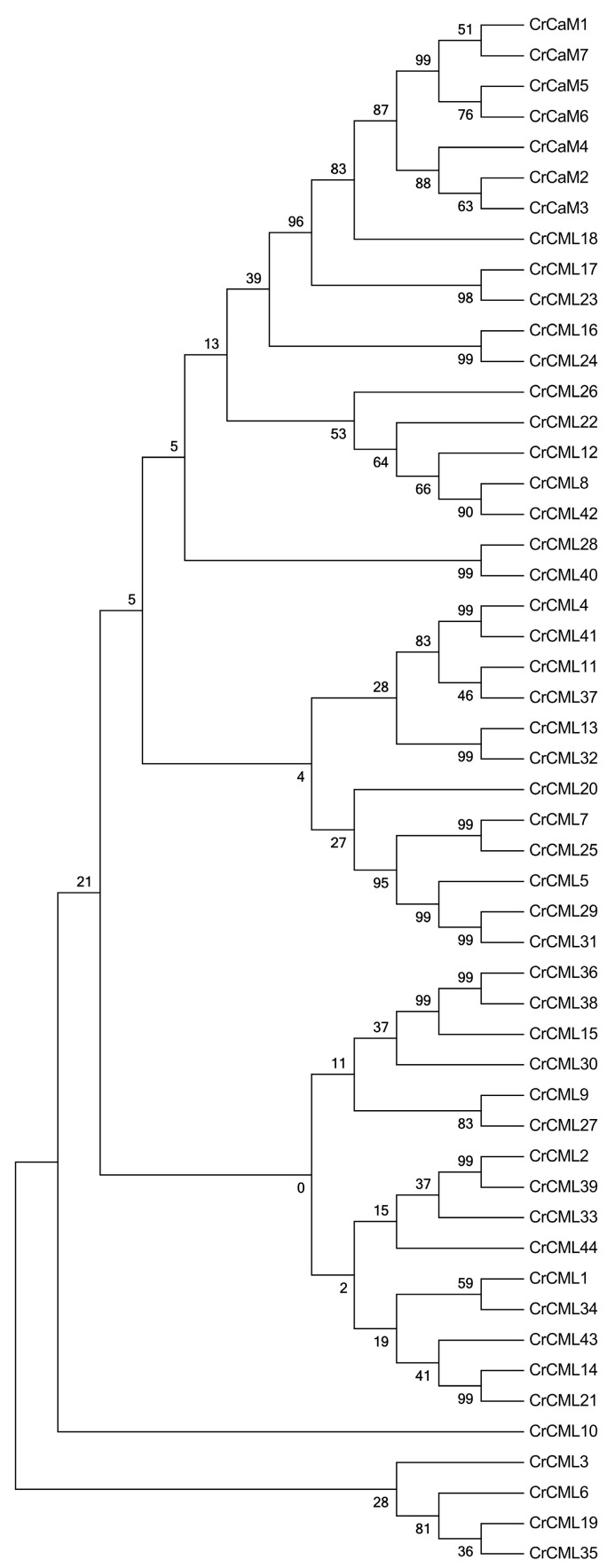
The phylogenetic tree of the CrCaMs/CrCMLs constructed using MEGA-X (version 10.1.8).

**Figure 6 ijms-25-11725-f006:**
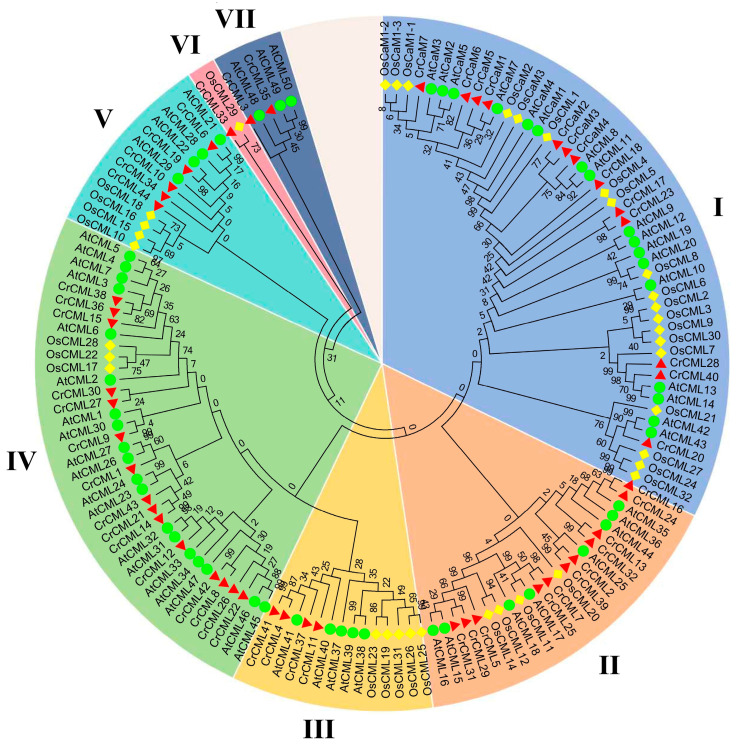
Phylogenetic relationships of the 51 CrCaMs/CrCMLs from *Canavalia rosea*, 57 AtCaMs/AtCMLs from *Arabidopsis thaliana*, and 37 OsCaMs/OsCMLs from *Oryza sativa*. The amino acid sequences of these 145 CaMs/CMLs from the three plant species were compared with a ClustalW alignment, and the phylogenetic tree was constructed in MEGA-X using the neighbor-joining method with 1000 bootstrap repetitions. The different branch colors represent different subgroups.

**Figure 7 ijms-25-11725-f007:**
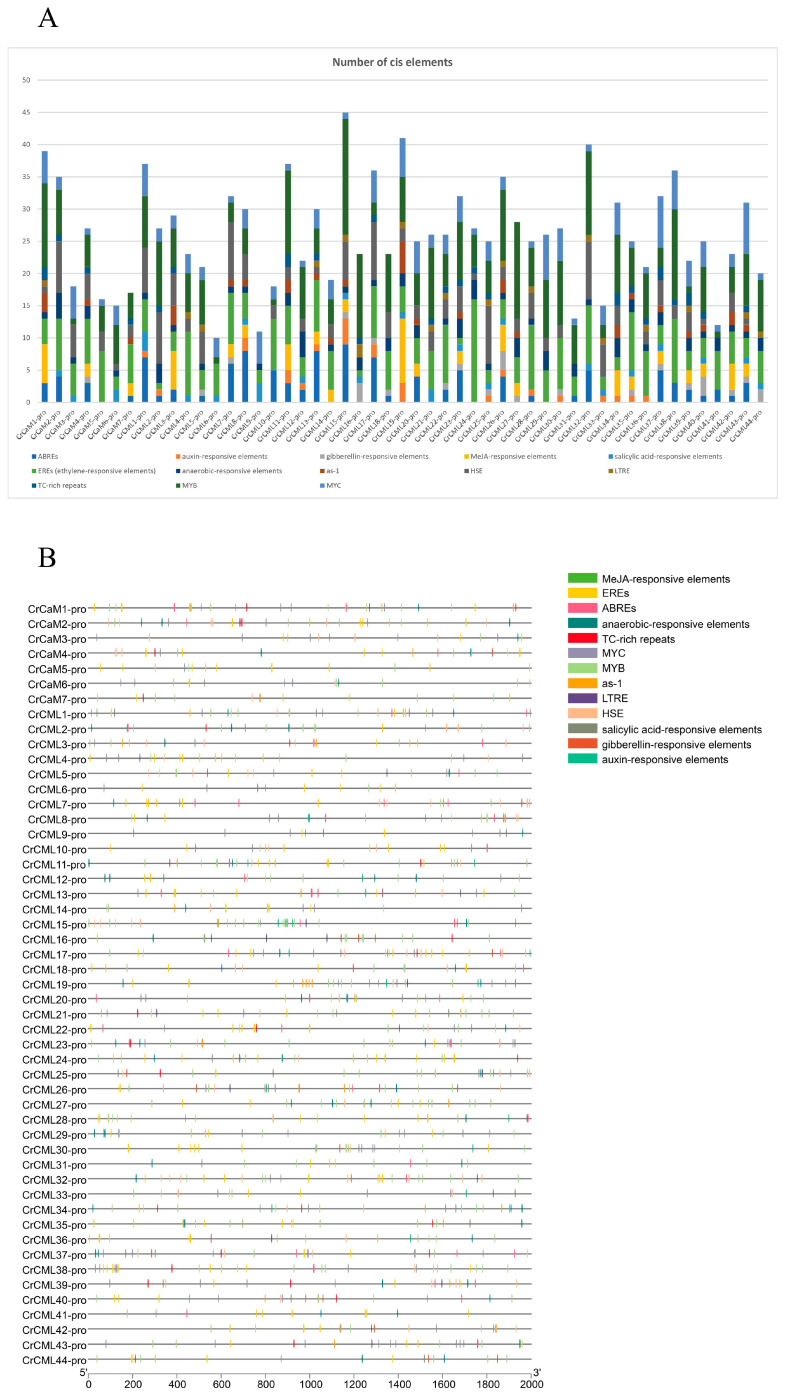
Statistics for the predicted *cis*-regulatory elements in the *CrCaM*s/*CrCML*s promoters (ATG upstream 2000 bp). (**A**) Summaries of the 13 *cis*-regulatory elements in the 51 *CrCaM*s/*CrCML*s promoter regions. (**B**) Distribution of these *cis*-regulatory elements in the 51 *CrCaM*s/*CrCML*s promoter regions. The elements are represented by different symbols. The scale bar represents 200 bp.

**Figure 8 ijms-25-11725-f008:**
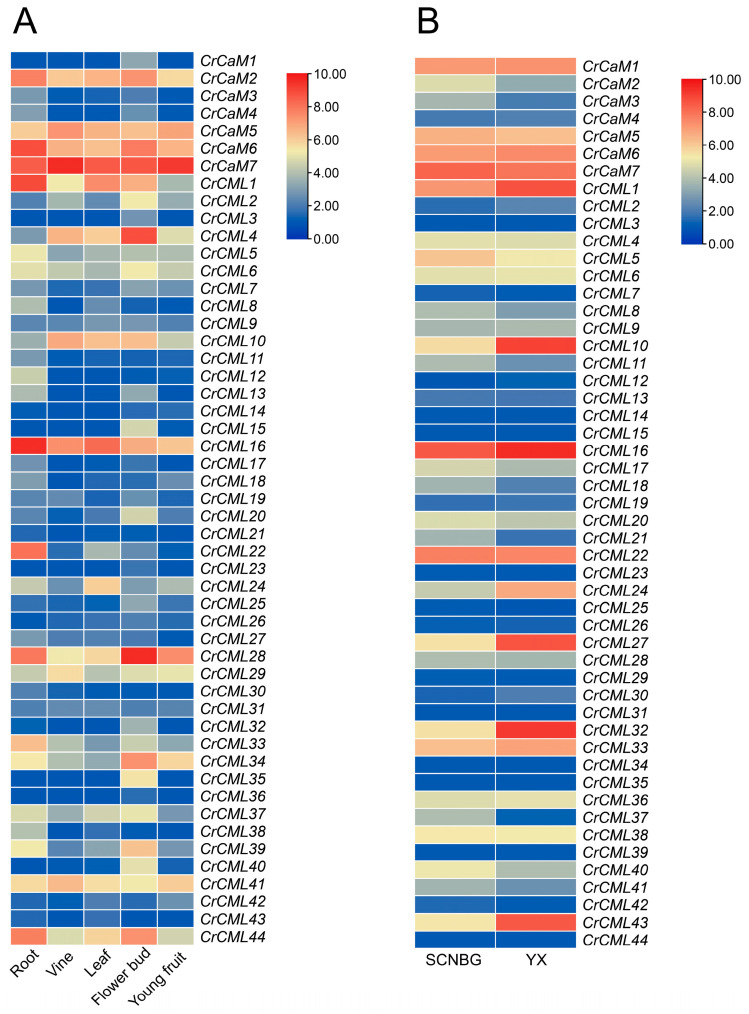
Heatmaps showing (**A**) the expression levels of the *CrCaM*s/*CrCML*s in the root, vine, leaf, flower bud, and young fruit of *Canavalia rosea* plants and (**B**) the expression differences in the *CrCaM*s/*CrCML*s in mature *C. rosea* leaves planted in the South China National Botanical Garden (SCNBG) and in Yongxing (YX) Island.

**Figure 9 ijms-25-11725-f009:**
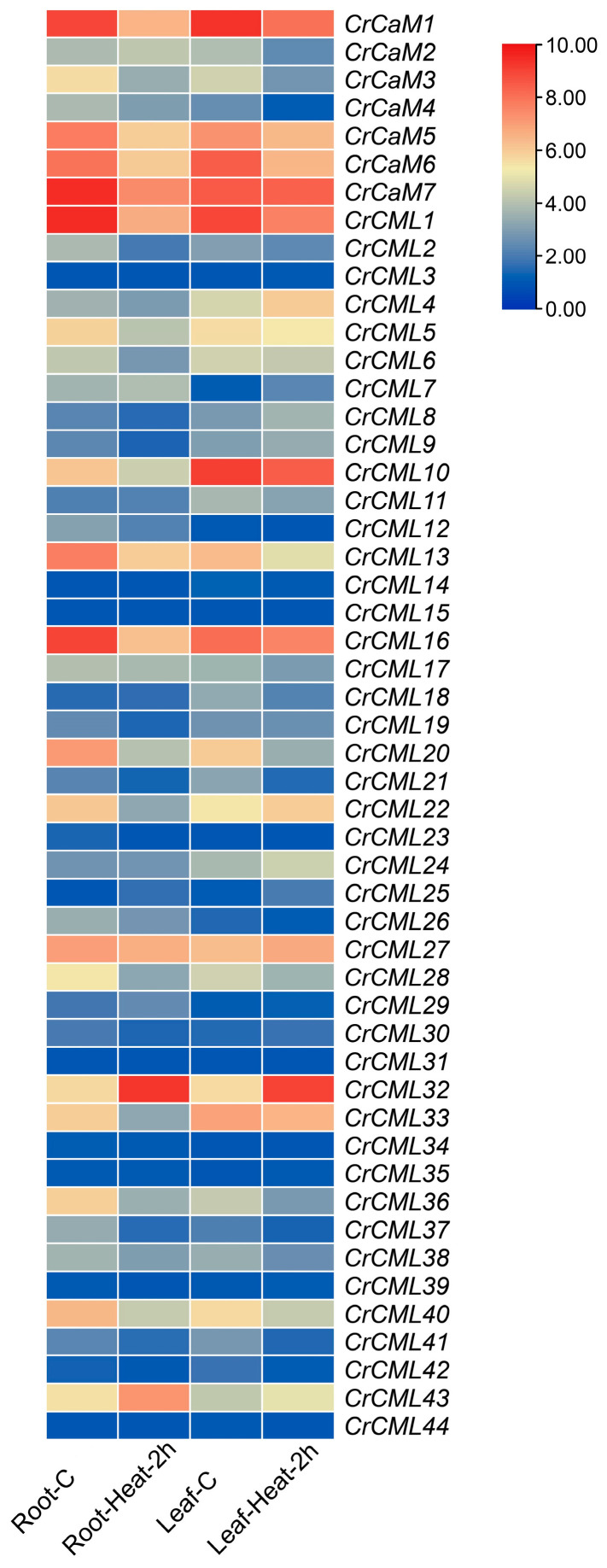
Heatmaps showing the expression levels of the RNA-Seq data for the *CrCaM*s/*CrCML*s in the root and leaf samples captured from the heat-shock-treated *Canavalia rosea* seedlings. The expression levels of each gene are shown in values of log2 (FPKM+1). Red denotes high expression levels and green denotes low expression levels.

**Figure 10 ijms-25-11725-f010:**
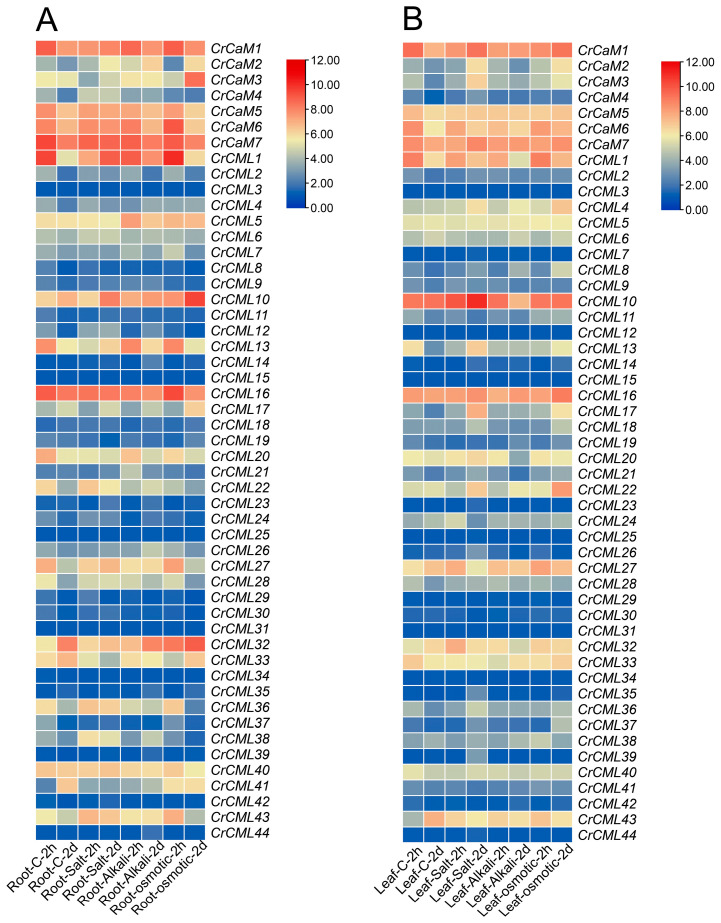
Heatmaps showing the expression changes in the RNA-Seq data for the *CrCaM*s/*CrCML*s under high salinity, alkaline, and high osmosis stresses. The expression levels of each gene are shown in values of log2 (FPKM+1). The expression differences of 51 *CrCaM*s/*CrCML*s in the root (**A**) and leaf (**B**) after the 2 h and 2 d abiotic stress challenges.

**Figure 11 ijms-25-11725-f011:**
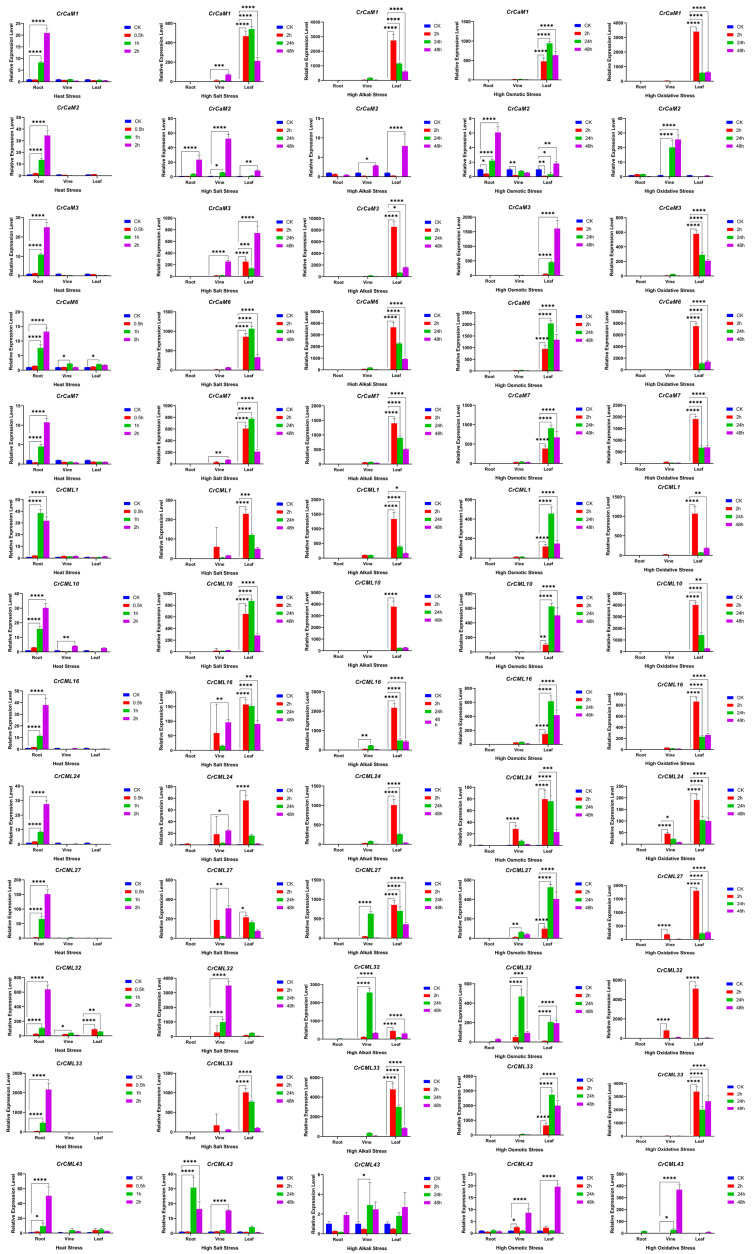
Quantitative RT-PCR detection of the expression levels of the thirteen *CrCaM*s/*CrCML*s responding to different abiotic stresses, including heat (45 °C), high salt (600 mM NaCl), alkaline (150 mM NaHCO_3_, pH 8.2), high osmotic stress (300 mM mannitol), and H_2_O_2_ (10 mM) oxidative stress. The relative expression values were calculated using the 2^−ΔCt^ method with the housekeeping gene, *CrEF-1α*, as a reference gene. The bars show the mean values ± SDs of n = 3–4 technical replicates. Asterisks indicate significant differences from the CK (control check, without stress, Student’s *t*-test, * *p* < 0.1, and ** *p* < 0.01, *** *p* < 0.001, and **** *p* < 0.0001).

**Figure 12 ijms-25-11725-f012:**
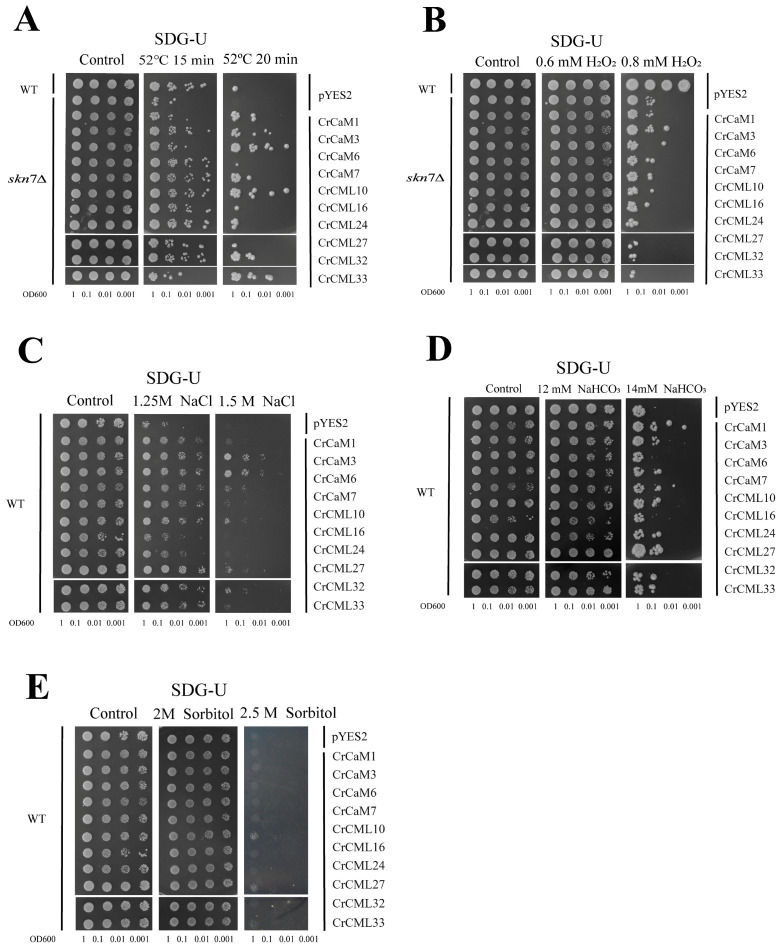
The functional confirmation in yeast strain WTs and *skn7Δ* by expressing ten *CrCaM*s/*CrCML*s. Yeast cultures (WT: 52 °C for 30 min for heat stress; *skn7Δ*: 52 °C for 15 min for heat stress; or WT without challenges) were adjusted to OD600 = one, and 2 μL of serial dilutions (ten-fold) were spotted on SDG-Ura medium plates supplied with specific challenge stressors. The plates were incubated for 2–5 days at 30 °C. Functional identification of ten *CrCaM*s/*CrCML*s in yeast using a heterologous expression assay. (**A**) The thermotolerance confirmation in the yeast mutant strain, *skn7Δ*; (**B**) H_2_O_2_ oxidative stress tolerance confirmation in the yeast mutant strain, *skn7Δ*; (**C**) high-salinity tolerance confirmation in the WT yeast on NaCl-surplus SDG-Ura medium plates; (**D**) high-alkaline tolerance confirmation in the WT yeast on NaHCO_3_-surplus SDG-Ura medium plates; and (**E**) the high-osmotic-stress tolerance confirmation in the WT yeast on sorbitol-surplus SDG-Ura medium plates.

**Table 1 ijms-25-11725-t001:** Nomenclature and subcellular localization of CrCaMs and CrCMLs identified from *Canavalia rosea* genome.

Name	Locus	Protein Length	Major Amino Acids (aa, %)	Mw (kDa)	PI	II	AI	GRAVY	Disordered aa (%)	WoLF_PSORT	Plant-PLoc
CrCaM1	03G007739	149	D(12.8%), E(12.8%), L(7.4%)	16.85	4.11	24.52	69.40	−0.619	32.89	nucl: 4, mito: 4, extr: 3, cyto: 2	Endoplasmic reticulum
CrCaM2	04G011067	171	E(14.0%), D(9.9%), L(8.8%)	19.68	4.05	38.15	84.33	−0.337	30.99	cyto: 6, chlo: 3, nucl: 3, plas: 1	Chloroplast
CrCaM3	04G011068	150	E(15.3%), D(12.0%), L(9.3%)	17.04	3.89	46.07	89.00	−0.331	25.33	cyto: 8, chlo: 2, extr: 1, E.R.: 1, cysk: 1	Nucleus
CrCaM4	04G013039	150	E(14.7%), D(12.7%), L(8.0%)	17.03	4.03	35.84	81.20	−0.487	31.33	chlo: 5, cyto: 4.5, cyto_nucl: 3.33333, extr: 3	Endoplasmic reticulum
CrCaM5	06G017463	190	D(11.1%), D(10.5%), L(10.0%)	21.59	4.14	33.12	80.58	−0.268	42.11	chlo: 6, plas: 2, nucl: 1.5, cysk_nucl: 1.5, cyto: 1, mito: 1, vacu: 1	Chloroplast
CrCaM6	07G020185	149	E(13.4%), D(12.1%), L(7.4%)	16.86	4.12	25.82	69.40	−0.619	32.89	nucl: 4, mito: 4, extr: 3, cyto: 2	Endoplasmic reticulum
CrCaM7	11G029359	149	D(12.8%), E(12.8%), A(7.4%), L(7.4%)	16.83	4.11	23.23	70.07	−0.602	32.89	cyto: 4, mito: 4, nucl: 3, extr: 2	Endoplasmic reticulum
CrCML1	01G000055	177	N(10.7%), D(10.7%), E(8.5%), G(8.5%)	19.60	4.23	39.80	59.49	−0.808	58.19	nucl: 6, chlo: 4, mito: 2, extr: 2	Nucleus
CrCML2	01G000134	187	D(8.2%), S(8.2%), E(9.1%), G(9.1%)	20.52	4.30	45.03	75.56	−0.396	55.61	chlo: 10, mito: 4	Nucleus
CrCML3	01G001229	81	E(12.3%), D(11.1%), L(11.1%)	9.21	4.62	41.03	85.43	−0.593	22.22	cyto: 5, nucl: 3.5, chlo: 3, cysk_nucl: 2.5, extr: 1	Cytoplasm
CrCML4	01G001582	191	S(10.5%), D(9.9%), G(8.4%)	21.71	5.12	49.75	56.65	−0.700	65.97	mito: 9, nucl: 4	Chloroplast
CrCML5	01G001986	160	L(11.2%), D(8.1%), G(8.1%)	17.62	4.21	35.31	83.56	−0.124	38.75	nucl: 5.5, nucl_plas: 4, cyto: 3.5, cyto_E.R.: 2.5, mito: 2, plas: 1.5	Nucleus
CrCML6	01G002018	288	E(11.1%), L(9.7%), F(8.7%)	33.26	4.97	42.59	76.56	−0.316	42.01	cyto: 5, chlo: 3, plas: 3, E.R.: 2	Chloroplast
CrCML7	01G002045	163	L(12.3%), A(11.0%), E(9.2%)	18.13	4.66	39.69	84.54	−0.315	43.56	plas: 5, nucl_plas: 5, nucl: 3, cyto: 2, mito: 2, chlo: 1	Nucleus
CrCML8	01G002566	179	L(11.2%), E(10.6%), N(7.8%)	20.41	4.37	46.73	87.65	−0.046	42.46	chlo: 11, cyto: 1, extr: 1	Chloroplast
CrCML9	01G002767	193	D(9.3%), K(9.3%), S(8.8%)	22.32	8.70	39.56	60.57	−0.627	60.10	nucl: 7.5, cyto_nucl: 4.5, chlo: 4, mito: 2	Cytoplasm
CrCML10	01G003442	155	G(14.2%), D(12.9%), L(7.7%)	16.86	4.81	31.39	66.71	−0.414	72.90	cyto: 6, mito: 4, chlo: 2, nucl: 1	Chloroplast
CrCML11	02G004181	197	D(10.7%), L(9.6%), S(9.6%)	21.89	4.55	41.87	72.79	−0.406	63.45	nucl: 7, cyto: 3, mito: 2, extr: 2	Cytoplasm
CrCML12	02G004464	187	G(10.7%), L(10.7%), E(10.2%)	20.98	4.69	36.33	78.13	−0.415	45.99	cyto: 11, nucl: 1, mito: 1	Cytoplasm
CrCML13	02G004981	139	L(14.4%), E(10.8%), D(9.4%)	16.00	4.34	27.74	80.58	−0.491	34.53	nucl: 5, mito: 5, chlo: 3	Cytoplasm
CrCML14	02G006679	214	L(9.8%), S(9.8%), D(9.3%)	24.15	5.18	32.02	76.50	−0.309	47.20	chlo: 11, nucl: 1, mito: 1	Cytoplasm
CrCML15	03G007802	152	D(11.8%), G(10.5%), E(9.9%)	17.03	4.32	15.81	73.09	−0.447	32.24	chlo: 4, extr: 4, nucl: 2, cyto: 2, mito: 2	Cytoplasm
CrCML16	03G007833	223	S(13.9%), D(9.4%), V(9.0%)	24.55	4.63	48.90	73.32	−0.526	60.54	chlo: 5, cyto: 5, nucl: 2, mito: 1	Chloroplast
CrCML17	03G008082	148	E(14.2%), D(10.1%), I(9.5%)	17.00	4.14	38.60	84.39	−0.443	27.03	cyto: 13	Chloroplast
CrCML18	03G009031	149	E(16.1%), D(10.7%), L(10.7%)	16.93	4.02	44.45	83.09	−0.448	38.26	cyto_nucl: 5.83333, chlo: 4, nucl: 4, cyto: 3.5, cyto_E.R.: 2.83333	Chloroplast
CrCML19	03G009705	220	L(10.5%), E(8.6%), I(8.6%)	25.58	5.69	27.67	86.00	−0.378	41.82	cyto: 10, mito: 2, chlo: 1	Chloroplast
CrCML20	03G010270	182	L(12.1%), D(8.8%), E(8.8%)	20.31	4.37	39.87	88.41	−0.234	41.21	mito: 7.5, chlo_mito: 7, chlo: 5.5	Nucleus
CrCML21	03G010292	206	L(10.2%), S(10.2%), D(9.2%)	23.42	5.47	33.93	79.95	−0.242	30.10	chlo: 10, extr: 2, cyto: 1	Chloroplast
CrCML22	03G010654	203	E(11.3%), F(10.8%), S(10.3%)	23.75	4.75	52.45	72.96	−0.467	38.92	nucl: 9, cyto: 2, extr: 2	Chloroplast
CrCML23	04G011122	148	E(13.5%), D(12.8%), L(8.8%)	17.08	4.05	24.54	88.85	−0.420	32.43	cyto: 6, cyto_nucl: 6, chlo: 4, extr: 1	Cytoplasm
CrCML24	04G011931	206	S(10.7%), A(9.7%), D(9.2%), G(9.2%)	22.30	4.62	36.41	71.55	−0.270	66.02	nucl: 7, mito: 4, chlo: 2	Chloroplast
CrCML25	04G012182	163	L(13.5%), E(9.8%), A(9.2%)	18.29	4.53	42.13	91.66	−0.279	30.06	nucl_plas: 5.5, nucl: 5, plas: 4, cyto: 2, mito: 2	Chloroplast
CrCML26	04G012874	84	E(15.5%), V(11.9%), L(8.3%)	9.87	4.38	59.01	92.62	−0.230	23.81	cyto: 5, extr: 5, nucl: 2, chlo: 1	Chloroplast
CrCML27	05G014580	172	K(11.6%), S(10.5%), E(9.3%), G(9.3%)	19.08	5.18	26.33	58.37	−0.505	47.09	cyto: 7, nucl: 3, chlo: 2, mito: 2	Nucleus
CrCML28	05G014903	145	E(11.0%), L(10.3%), D(9.0%)	16.53	4.86	31.69	84.76	−0.403	35.86	nucl: 4, cyto: 3, plas: 2, chlo: 1, mito: 1, extr: 1, cysk: 1	Chloroplast
CrCML29	05G015286	160	L(13.8%), A(8.8%), G(8.8%)	17.49	4.41	27.70	95.75	−0.022	40.00	plas: 5, nucl_plas: 4.5, cyto: 4, nucl: 2, chlo: 1, mito: 1	Chloroplast
CrCML30	05G015406	224	E(12.1%), L(9.8%), K(9.8%)	25.17	4.83	44.01	88.75	−0.214	50.00	chlo: 9, extr: 3, nucl: 1	Cytoplasm
CrCML31	05G017030	160	L(13.8%), A(9.4%), D(8.8%), G(8.8%)	17.59	4.30	32.93	90.31	0.057	42.50	chlo: 10, mito: 2, nucl: 1	Chloroplast
CrCML32	06G018151	159	L(15.1%), D(10.7%), E(10.7%)	18.21	4.31	46.05	86.42	−0.359	43.40	nucl: 5, cyto: 5, chlo: 2, mito: 1	Cytoplasm
CrCML33	06G018267	157	L(17.2%), E(12.1%), G(10.2%)	16.96	4.10	25.21	108.09	−0.061	33.76	cyto: 10.5, cyto_E.R.: 6.33333, E.R._vacu: 1.33333	Nucleus
CrCML34	06G018865	84	D(11.9%), E(10.7%), A(9.5%)	9.31	4.21	20.60	67.38	−0.568	26.19	nucl: 5, cyto: 3, mito: 3, chlo: 1, plas: 1	Mitochondrion
CrCML35	06G019154	98	G(9.2%), A(8.2%), K(8.2%)	11.20	8.79	20.00	81.63	−0.587	61.22	mito: 8.5, chlo_mito: 6, chlo: 2.5, nucl: 1.5, cyto_nucl: 1.5	Chloroplast
CrCML36	08G022893	150	D(12.7%), G(10.7%), E(10.0%)	16.99	4.34	26.75	72.07	−0.516	29.33	nucl: 4, chlo: 3, mito: 3, cyto: 2, extr: 2	Cytoplasm
CrCML37	09G023581	141	E(14.9%), L(12.1%), G(9.9%)	16.14	4.34	46.10	75.39	−0.365	29.08	nucl: 6, chlo: 4, cyto: 2, mito: 1	Chloroplast
CrCML38	09G025115	218	D(9.6%), L(9.6%), I(8.7%)	24.88	4.74	41.15	88.03	−0.259	50.00	chlo: 12, mito: 1	Nucleus
CrCML39	10G025518	191	S(12.0%), E(11.5%), D(8.9%)	21.41	4.58	46.26	69.37	−0.468	46.07	chlo: 7, mito: 5, nucl: 2	Nucleus
CrCML40	10G026298	149	D(11.4%), L(9.4%), E(7.4%)	16.89	4.70	34.32	80.54	−0.436	59.06	cyto: 9, cyto_nucl: 8, nucl: 3	Cytoplasm
CrCML41	10G026655	190	S(12.6%), D(10.5%), L(10.5%)	21.90	4.77	51.14	70.32	−0.670	61.05	mito: 7.5, chlo_mito: 5.5, nucl: 4, chlo: 2.5	Chloroplast
CrCML42	10G027366	165	E(11.5%), F(9.1%), V(8.5%)	19.30	4.82	45.76	83.15	−0.259	24.85	extr: 4, vacu: 3, E.R.: 3, golg: 2, chlo: 1	Chloroplast
CrCML43	11G028665	210	S(12.9%), K(10.0%), G(8.1%), L(8.1%)	23.37	7.63	35.58	78.86	−0.304	47.14	nucl: 11, cyto: 3	Nucleus
CrCML44	11G028666	84	A(10.7%), D(10.7%), E(10.7%)	9.31	4.37	14.27	74.40	−0.425	35.71	cyto: 8, nucl: 2, mito: 2, chlo: 1	Peroxisome

MW: molecular weight; PI: isoelectric point; II: instability index; AI: aliphatic index; GRAVY: grand average of hydropathicity. The molecular weight and isoelectric points of predicted CrCaMs/CrCMLs were detected using the ExPASy proteomics server (https://web.expasy.org/protparam/, accessed on 10 April 2024). The TMHMM Server 2.0 program (http://www.cbs.dtu.dk/services/TMHMM/) and the Protein Fold Recognition Server tool (PHYRE^2^, http://www.sbg.bio.ic.ac.uk/phyre2/html/page.cgi?id= index, accessed on 10 April 2024) were used to predict the transmembrane helices and the 3D prediction of CrCaMs/CrCMLs. For the subcellular localization prediction, the online program WoLF_PSORT (https://www.genscript.com/wolf-psort.html, accessed on 10 April 2024) was used.

**Table 2 ijms-25-11725-t002:** Ka/Ks analysis and duplicated type calculation for *CrCaM* and *CrCML* genes.

Duplicated Pair	Duplicate Type	Ka	Ks	Ka/Ks	*p*-Value (Fisher)	Positive Selection
*CrCaM3*-*CrCaM4*	Segmental	0.12532	0.542019	0.231209	6.07 × 10^−9^	No
*CrCaM5*-*CrCaM7*	Segmental	0.00565853	0.507362	0.0111528	7.7 × 10^−24^	No
*CrCML2*-*CrCML39*	Segmental	0.224798	1.46238	0.153721	4.27 × 10^−20^	No
*CrCML4*-*CrCML41*	Segmental	0.147676	0.735966	0.200656	3.32 × 10^−14^	No
*CrCML8*-*CrCML42*	Segmental	0.280668	0.761875	0.368392	5.48 × 10^−6^	No
*CrCML13*-*CrCML32*	Segmental	0.200534	0.803216	0.249664	1.16 × 10^−8^	No
*CrCML14*-*CrCML21*	Segmental	0.199284	0.538055	0.370378	1.25 × 10^−6^	No
*CrCML16*-*CrCML24*	Segmental	0.250644	0.787603	0.318237	3.62 × 10^−10^	No
*CrCML17*-*CrCML23*	Segmental	0.157841	0.707805	0.223001	2.98 × 10^−10^	No
*CrCML18*-*CrCML26*	Segmental	0.137486	0.583507	0.235619	1.3 × 10^−8^	No
*CrCML28*-*CrCML40*	Segmental	0.0930564	0.609897	0.152577	2.73 × 10^−13^	No
*CrCML29*-*CrCML31*	Segmental	0.0905119	0.792094	0.114269	8.83 × 10^−20^	No
*CrCML34*-*CrCML44*	Segmental	0.0645313	0.529471	0.121879	1.74 × 10^−8^	No
*CrCML36*-*CrCML38*	Segmental	0.112235	1.05912	0.105971	1.19 × 10^−19^	No
*CrCaM2*-*CrCaM3*	Tandem	\	\	\	\	\
*CrCML43*-*CrCML44*	Tandem	\	\	\	\	\

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
