# Peer review of "Identifying Calmodulin and Calmodulin-like Protein Members in Canavalia rosea and Exploring Their Potential Roles in Abiotic Stress Tolerance"

_ijms, 2024, doi:10.3390/ijms252111725_

Round 1
Reviewer 1 Report
Comments and Suggestions for Authors
Dear Authors,
Your manuscript titled „Genome-wide characterization and transcriptional analyses of the calmodulin and calmodulin-like protein family in Canavalia rosea provide insight into their potential roles in abiotic stress tolerance” contains valuable results. Nevertheless, I have found some imperfections, which (in my opinion) should be corrected or at least clarified before an eventual publication. I have listed them below:
1. I suggest to shorten title of manuscript.
2. In my opinion in chapter Introduction brief characteristics of study species should be added. Such characteristics should contain information about life form, life span, range and habitat afiliation of Canavalia rosea. Moreover the current state of knowledge on adaptation to abiotic stress in Canavalia rosea should be presented. Also, the choice of aforementioned species as object of investigations should be stronger justified.
3. Chapter Introduction should be ended by listing the aims of investigations. Lines 84-89 should be moved into chapter Material and methods.
4. Figures 1-4 and 6-11 are hardly legible. Their quality should be improved.
5. Please add the chapter Conclusions.
6. Please, look into following publications. Perhaps, some of them would be useful in manuscript corrections:
· Lin, R., Zheng, J., Pu, L., Wang, Z., Mei, Q., Zhang, M., & Jian, S. (2021). Genome-wide identification and expression analysis of aquaporin family in Canavalia rosea and their roles in the adaptation to saline-alkaline soils and drought stress. BMC plant biology, 21, 1-23.
· Mohajer, S., Taha, R. M., Mohamed, N., & Razak, U. N. A. (2017). Baybean (Canavalia rosea (Sw.) DC.); organogenesis, morphological and anatomical studies. Gayana Bot, 74, 1.
· Mendoza-González, G., Martínez, M. L., & Lithgow, D. (2014). Biological flora of coastal dunes and wetlands: Canavalia rosea (Sw.) DC. Journal of Coastal Research, 30(4), 697-713.
Author Response
Dear Authors,
Your manuscript titled “Genome-wide characterization and transcriptional analyses of the calmodulin and calmodulin-like protein family in Canavalia rosea provide insight into their potential roles in abiotic stress tolerance” contains valuable results. Nevertheless, I have found some imperfections, which (in my opinion) should be corrected or at least clarified before an eventual publication. I have listed them below:
- I suggest to shorten title of manuscript.
Response: We have simplified the title as: “Identification of calmodulin and calmodulin-like protein members in Canavalia rosea and exploring their potential roles in abiotic stress tolerance”.
- In my opinion in chapter Introduction brief characteristics of study species should be added. Such characteristics should contain information about life form, life span, range and habitat afiliation of Canavalia rosea. Moreover the current state of knowledge on adaptation to abiotic stress in Canavalia roseashould be presented. Also, the choice of aforementioned species as object of investigations should be stronger justified.
Response: Thanks for your reminder. We have added the expression in introduction part as: “C. rosea is a common vine species in south China coastal wetland with excellent tolerance to barren soil and salinity/alkaline, and fast growth ability for horticultural engineering operation. Moreover, due to its seeds’ nutritional and medicinal values, C. rosea has also become an important wild plant resource with latent economic value and the ecology significance [16, 17]. ”.
- Chapter Introduction should be ended by listing the aims of investigations. Lines 84-89 should be moved into chapter Material and methods.
Response: Thanks for your suggestion. We have adjusted the “Introduction” section and emphasized the aims of this study. Also, we removed the original Lines 84–89 to “Discussion” section.
- Figures 1-4 and 6-11 are hardly legible. Their quality should be improved.
Response: Thanks for your suggestion. We have replaced these figures with more higher resolution images.
- Please add the chapter Conclusions.
Response: The conclusion section has been added.
- Please, look into following publications. Perhaps, some of them would be useful in manuscript corrections:
Response: Thanks for your reminder.
- Lin, R., Zheng, J., Pu, L., Wang, Z., Mei, Q., Zhang, M., & Jian, S. (2021). Genome-wide identification and expression analysis of aquaporin family in Canavalia rosea and their roles in the adaptation to saline-alkaline soils and drought stress. BMC plant biology, 21, 1-23.
- Mohajer, S., Taha, R. M., Mohamed, N., & Razak, U. N. A. (2017). Baybean (Canavalia rosea (Sw.) DC.); organogenesis, morphological and anatomical studies. Gayana Bot, 74, 1.
- Mendoza-González, G., Martínez, M. L., & Lithgow, D. (2014). Biological flora of coastal dunes and wetlands: Canavalia rosea (Sw.) DC. Journal of Coastal Research, 30(4), 697-713.
Submission Date
29 September 2024
Date of this review
07 Oct 2024 07:21:02

Reviewer 2 Report
Comments and Suggestions for Authors
This study provides a foundation for understanding the roles of CaM and CML genes in plant stress resistance, particularly in halophytes like C. rosea. It offers insights into the natural ecological adaptability of this species and provides a theoretical basis for further research on the physiological functions of CrCaMs and CrCMLs in response to multiple abiotic stresses.
Overall, the purpose of the study is suitable for publication, and the authors provide substantial results to support their hypothesis. However, the presentation of figures requires significant revision based on the following comments:
1. Figure 1: Chromosomal locations should be displayed in a single row.
2. Figure 2: CrCaMs and CrCMLs should be distinguished using different colors.
3. Motif consensus figures: Increase the size to improve readability.
4. Figures 3 and 4: Increase font sizes to enhance readability.
5. Figures 5 and 6: Phylogenetic trees should be generated using the maximum likelihood method instead of the neighbor-joining method.
6. Figure 7: Increase font sizes, remove frames, and reposition labels A and B appropriately (not at the bottom).
7. Figures 8 and 10: Remove frames and reposition labels A and B appropriately (not at the bottom).
8. Figure 11: Reorganize and enlarge the qRT-PCR results to improve readability.
9. Figure 12: Reorganize the results for better presentation.
These revisions will significantly improve the overall quality and clarity of the figures, making the study's findings more accessible to readers.
Author Response
This study provides a foundation for understanding the roles of CaM and CML genes in plant stress resistance, particularly in halophytes like C. rosea. It offers insights into the natural ecological adaptability of this species and provides a theoretical basis for further research on the physiological functions of CrCaMs and CrCMLs in response to multiple abiotic stresses.
Overall, the purpose of the study is suitable for publication, and the authors provide substantial results to support their hypothesis. However, the presentation of figures requires significant revision based on the following comments:
- Figure 1: Chromosomal locations should be displayed in a single row.
Response: Thanks. We have rearranged the chromosomes and made them being in single row.
- Figure 2: CrCaMs and CrCMLs should be distinguished using different colors.
Response: The two pairs of CrCaMs (CrCaM3 and CrCaM4, CrCaM5 and CrCaM7) in Figure 2 have been distinguished with yellow background and red typeface.
- Motif consensus figures: Increase the size to improve readability.
Response: Done.
- Figures 3 and 4: Increase font sizes to enhance readability.
Response: Done.
- Figures 5 and 6: Phylogenetic trees should be generated using themaximum likelihood methodinstead of the neighbor-joining method.
Response: Actually, We found in most of researches, their phylogenetic trees were constructed with Neighbor-Joining (NJ) method, not maximum likelihood method, and the bootstrap of NJ method is high enough (1000), so here we chose the NJ method to perform the evolutionary analysis of three species CaM and CML proteins, including Arabidopsis, rice, and our target plant, C. rosea.
- Figure 7: Increase font sizes, remove frames, and reposition labels A and B appropriately (not at the bottom).
Response: We have rearranged or replaced the Figures 7 to 11, to make them more clear and more readable.
- Figures 8 and 10: Remove frames and reposition labels A and B appropriately (not at the bottom).
Response: Done.
- Figure 11: Reorganize and enlarge the qRT-PCR results to improve readability.
Response: Done.
- Figure 12: Reorganize the results for better presentation.
Response: All figures were replaced with higher quality figures, and pictures were also re-organized (Figures 8, 10, and 12). Also some grammatical errors and less rigorous expression have been adjusted and improved.
These revisions will significantly improve the overall quality and clarity of the figures, making the study’s findings more accessible to readers.
Submission Date
29 September 2024
Date of this review
06 Oct 2024 10:22:34

Round 2
Reviewer 1 Report
Comments and Suggestions for Authors
Dear Authors,
Thank You for implementing my suggestions in the text. I do not have any further remarks.
Author Response
I do not have any further remarks.
Response: Thanks for your affirmation and patience.
Reviewer 2 Report
Comments and Suggestions for Authors
I have reviewed the revised manuscript.
Most revisions have been nicely implemented as per the reviewer's recommendations.
However, the quality of some figures, specifically Figures 1, 3, 4, 7, 11, and 12, is very poor.
The authors should provide high-quality images for these figures.
I recommend that after converting the manuscript to PDF, the authors carefully check the images themselves.
This will help them understand why I am requesting high-quality images.
Additionally, please provide a clean version of the manuscript.
Once these issues are addressed, the manuscript can be considered acceptable for publication.
Congratulations on your excellent work.
Author Response
The quality of some figures, specifically Figures 1, 3, 4, 7, 11, and 12, is very poor. The authors should provide high-quality images for these figures.
Response: Figure 1 has been adjusted as two rows, that wound be seemed to be more convenient to check the genes’ location information. Figures 3 and 4 have been redone and rearranged to increase the clarity. The figures 7, 11, and 12 have been replaced with high-quality images.
I recommend that after converting the manuscript to PDF, the authors carefully check the images themselves. This will help them understand why I am requesting high-quality images.
Response: Thanks for your reminder. We do find some minor mistakes and correct them, both in text and in figures.
Additionally, please provide a clean version of the manuscript.
Response: The clean version have been uploaded.